Identification of MHCII variants associated with chlamydial disease in the koala (Phascolarctos cinereus)

Lau Quintin
Griffith Joanna E.
Higgins Damien P. damien.higgins@sydney.edu.au
Faculty of Veterinary Science, The University of Sydney , NSW , Australia
Yang Xiang-Jiao
Electronic publication date: 2014 Jun 19
Publication date: 2014
Volume: 2
Electronic Location ID: e443
Received 2014 Mar 21; Accepted 2014 Jun 2
Copyright: © 2014 Lau et al.
Copyright year: 2014
Copyright holder: Lau et al.
License: This is an open access article distributed under the terms of the Creative Commons Attribution License, which permits unrestricted use, distribution, reproduction and adaptation in any medium and for any purpose provided that it is properly attributed. For attribution, the original author(s), title, publication source (PeerJ) and either DOI or URL of the article must be cited.
License URL: https://creativecommons.org/licenses/by/4.0/

Keywords: Marsupial, Immunology, MHC, Chlamydia, Disease, Major histocompatibility complex, Koala, Heat shock proteins

Funding: Hermon Slade Foundation HSF0608 This research work was funded by the Hermon Slade Foundation (HSF0608). The funders had no role in study design, data collection and analysis, decision to publish, or preparation of the manuscript.

==============================
Chlamydiosis, the most common infectious disease in koalas, can cause chronic urogenital tract fibrosis and infertility. High titres of serum immunoglobulin G against 10 kDa and 60 kDa chlamydial heat-shock proteins (c-hsp10 and c-hsp60) are associated with fibrous occlusion of the koala uterus and uterine tube. Murine and human studies have identified associations between specific major histocompatibility complex class II (MHCII) alleles or genotypes, and higher c-hsp 60 antibody levels or chlamydia-associated disease and infertility. In this study, we characterised partial MHCII DAB and DBB genes in female koalas (n = 94) from a single geographic population, and investigated associations among antibody responses to c-hsp60 quantified by ELISA, susceptibility to chlamydial infection, or age. The identification of three candidate MHCII variants provides additional support for the functional role of MHCII in the koala, and will inform more focused future studies. This is the first study to investigate an association between MHC genes with chlamydial pathogenesis in a non-model, free-ranging species.

Introduction

Chlamydiosis, caused predominantly by Chlamydia pecorum and C. pneumoniae, is the most common infectious disease in koalas, causing proliferative conjunctivitis and/or chronic, fibrotic disease of the urogenital tract, which can lead to infertility and death (Griffith et al., 2013; Obendorf, 1983). Heat shock proteins (hsp) are highly conserved polypeptides produced under stress conditions to preserve cellular activity, and are upregulated by Chlamydiae in their persistent intermediate forms, which are induced by IFNγ in sub-lethal Th1 immune responses (Beatty, Byrne & Morrison, 1994). Evidence suggests that most pathological damage from chlamydial infections results from type IV delayed-type hypersensitivity to chlamydial hsps 10 and 60, in a deleterious and non-protective response (Lichtenwalner et al., 2004; Morrison, 1991). In humans, high serum immunoglobulin G (IgG) titres against 10 kDa and 60 kDa chlamydial heat shock proteins (c-hsp10 and c-hsp60) are associated with chronic infection, salpingeal fibrosis, tubal infertility and ectopic pregnancy (Betsou, Sueur & Orfila, 1999; LaVerda et al., 2000; Toye et al., 1993; Zhong & Brunham, 1992); in Chlamydia-infected koalas, high anti c-hsp10 and c-hsp60 titres are also associated with fibrous occlusion of the uterus and uterine tube (Higgins, Hemsley & Canfield, 2005).

Chlamydial hsp60 antibody responses and Chlamydia-associated pathology could be genetically predisposed: in mice, greater c-hsp60 antibody responses are associated with alleles of genes in or near IA genes within the major histocompatibility complex (MHC) (Zhong & Brunham, 1992). The murine IA genes are highly polymorphic MHCII genes that encode for membrane-bound molecules that bind specific exogenous antigens and present them to T lymphocytes (Balakrishnan & Adams, 1995; Kalish, 1995). These are orthologous to the human HLA-DQ α and β genes and, similarly, two human class II alleles, DQA1*0401 and DQB1*0402 (DQ4 phenotype), are associated with increased prevalence and magnitude of c-hsp60 antibody titres (Gaur et al., 1999), and a class II haplotype DR8 DQ4 is associated with presence of anti c-hsp10 antibodies (Betsou, Sueur & Orfila, 1999). A number of studies of Chlamydia trachomatis infection in women have also identified associations between HLA (MHCII) genotypes and Chlamydia-associated pelvic inflammatory disease, cervicitis and tubal infertility (Cohen et al., 2003; Cohen et al., 2000; Ness et al., 2004).

Variation in MHC genes is widely considered important for resistance against pathogens and, therefore, fitness and long-term survival, based on the important role that the MHC has in the host immune system, with a growing number of studies identifying associations between MHC diversity, or specific alleles/variants, and disease or survival in wildlife species (Sommer, 2005). Examples include: specific MHC alleles of wild Soay sheep populations associated with parasite resistance and juvenile survival (Paterson, Wilson & Pemberton, 1998); in water voles and red junglefowl, heterozygote and homozygote individuals possessing a specific MHCII allele have higher survival rates than homozygotes of another allele (Oliver & Piertney, 2012; Worley et al., 2010); and increased juvenile survival in Seychelles warblers resulting from increased MHC diversity and a specific allele (Brouwer et al., 2010).

No marsupial orthologs to the MHC genes associated with human c-hsp60 antibody levels have been identified, due to the different MHC class II gene families of marsupials (Belov, Lam & Colgan, 2004). Partial koala MHCII genes have been characterised (Jobbins et al., 2012; Lau et al., 2013), and variation across Australia described (Lau et al., 2014). Though MHCII is clearly functional in the koala immune system, based on its conservation of genetic structure relative to other species (Lau et al., 2013) and demonstration of expression and up-regulation in koala lymphocytes and tissues (Canfield, Hemsley & Connolly, 1996; Lau, Canfield & Higgins, 2012), associations between koala MHCII and disease have not yet been investigated, but the low genetic diversity of some koala populations (Lau et al., 2014) makes this an area worthy of study. To open investigation of the relevance of MHCII diversity to chlamydial disease in the koala, this study makes use of archived samples and data from hospital admissions from a wild Chlamydia-exposed koala population to identify MHCII DA and DB β (DAB and DBB, respectively) variants as candidates for more targeted investigation in prospective studies. Specifically, we aim to identify if there are any associations between MHCII variants and antibody responses to c-hsp60; susceptibility to chlamydial infection; or age and, therefore, likelihood of survival.

Methods

Ethics statement

All samples using in this study were collected in a research project approved by the University of Sydney Animal Ethics Committee (permit number AEC N00/5-2009/1/4829). The project was funded by the Hermon Slade Foundation.

Study population

This study utilised archived samples and clinical records from female koalas (n = 94) from the Port Macquarie and Hastings River district on the mid-north coast of New South Wales, Australia. The samples were collected for clinical assessment of koalas following admission to the Koala Hospital, Port Macquarie, between 2005 and 2011. Blood samples were collected into serum, EDTA or heparin tubes and, following centrifugation, serum or plasma was aspirated and surplus blood cells, plasma and serum were stored at −20 °C. In addition, urogenital and conjunctival swabs were collected for chlamydial diagnosis (n = 37) and stored at −20 °C. Some koalas, which were euthanased for humane reasons under Koala Hospital operating guidelines (n = 47), were subjected to necropsy and tissues collected into 10% buffered formalin for histopathological confirmation for chlamydial disease and/or stored frozen at −20 °C. DNA samples were extracted from swabs, blood cells, or liver, using the DNeasy Blood and Tissue kit and protocol (Qiagen, Doncaster, Australia) and then stored at −20 °C.

Categorisation of koalas

Based on clinical and necropsy records, koalas were categorised into three health groups. Animals with urogenital swabs positive by polymerase chain reaction (PCR) amplification of ompA gene of C. pecorum following Higgins et al. (2011), or presence of clinical signs consistent with chlamydial disease, such as cystitis, rump staining, bladder thickening or ovarian bursal cysts, were classified “past infected” (n = 63). Remaining animals were classed as “healthy” (n = 16) or, where no clinical data were available, “unknown” (n = 15). Koalas of “unknown” health were included only in calculations investigating associations between MHCII genotypes and chlamydial hsp-60 antibody levels, and between MHCII genotype and age. The natural lifespan of female and male koalas is over 15 and 12 years of age, respectively (Martin & Handasyde, 1999). As young koalas are more likely to present to wildlife rehabilitators as a result of misadventure and trauma during dispersal, while older koalas more commonly present for diseases such as chlamydiosis (Griffith et al., 2013), koalas were divided into four age classes in order to assess if age was a confounding factor for antibody titres: young adult, AY, two to five years old (n = 15); mature adult, AM, five to ten years old (n = 33); aged adult, AA, over ten years old (n = 25); and age unknown AU (n = 21) based on an aging system utilising tooth wear class (Martin, 1981).

MHCII genotyping of koalas

Following Lau et al. (2013), koala-specific DAB and DBB exon 2 sequences were amplified in all koalas by PCR, using the primers DABEx2F: 5′-ATGCCCCAAAGCACTTCAC-3′, DABEx2R: 5′-CGCACTRAGAAGGGCTCA-3 and DBBEx2F: 5′-AGGGACATCCCAGAGGATTTCG-3′, DBBEx2R: 5′-TCTTCTGTCCACCGCGAAGG-3′. Forward primers were 5′-phosphorylated for enzymatic digestion of the amplicon forward strand prior to one-strand conformation polymorphism (OSCP) analysis. The PCR amplifications were carried out in 25 µl reactions with 10–30 ng of DNA, 0.32 µM each primer (Sigma-Aldrich, Sydney, Australia), 1 × HotStarTaq DNA Polymerase PCR Buffer, 1 mM MgCl2, 0.2 mM dNTPs, and 0.5 units of HotStarTaq DNA Polymerase (Qiagen). Cycle conditions were: initial activation at 95 °C for 15 min, followed by 35 cycles of 40 s denaturation at 95 °C, 40 s annealing at 57 °C or 54 °C (for DAB and DBB, respectively), and 45 s extension at 72 °C, and a final extension at 72 °C for 10 min.

For OSCP analysis, the forward strand of PCR products was digested with Lambda exonuclease (New England Biolabs, Ipswich, MA, USA) at 37 °C for 45 min, and the reverse strand was subjected to electrophoresis in a 5% or 10% acrylamide gel (DAB and DBB, respectively) at 30 W for 5 h at 4 °C (Lau et al., 2013). Individuals with “genotype patterns” that were identical to reference animals established by Lau et al. (2013) were considered to have those genotypes, as these OSCP patterns were demonstrated to have high discriminatory power in that study. Any new “genotype patterns” were referred to as novel and were characterised further to identify their constituent variants using direct sequencing and molecular cloning (Lau et al., 2013). DAB genotype was unavailable for one individual due to limited archived sample, and was accounted for in all statistical analyses.

ELISA for quantification of chlamydial hsp60-specific antibodies

Following Higgins, Hemsley & Canfield (2005), a three-stage indirect ELISA was used for quantification of c-hsp60 antibodies. Standard sera comprised of ten doubling dilutions of a patient plasma sample with high c-hsp60 antibody titres and recombinant c-hsp60 peptide antigens were those used by Higgins, Hemsley & Canfield (2005) and LaVerda et al. (2000). Wells of a 96-well plate (Immulon 2HB; Thermo Labsystems, Franklin, MA, USA) were allocated as test (antigen and patient sera), control (patient sera, no antigen) and blank (no antigen, no sera), and incubated with 0.1 µg recombinant antigen or no antigen, respectively, in 100 µl bicarbonate buffer (pH 9.6) at 4 °C for 28 h. Wells were then washed four times with TPBS (1 × PBS pH 7.6 with 0.01% Tween 20) and then incubated on a 37 °C water bath, with 200 µl 1% casein (Sigma, Castle Hill, Australia) in PBS (120 min), followed by 100 µl patient sera diluted 1/100 in TPBS, or 100 µl standard sera or 100 µl TPBS for blank wells (60 min), followed by 100 µl rabbit anti-kangaroo IgG (Bethyl Laboratories, Inc., Montgomery, TX, USA) 1/50 in TPBS (60 min), then 100 µl alkaline phosphatise-conjugated donkey anti-rabbit Ig (Chemicon International, Inc., Billerica, MA, USA) diluted 1/1200 in TPBS (60 min). Between incubations, wells were washed four times with TPBS and, after the final incubation, with TBS (tris-buffered saline). To create a signal, wells were incubated with 100 µl p-nitrophenylphosphate (pNpp; Bio-Rad, Hercules, CA, USA) at room temperature and the reaction was stopped at 6 min using 50 µl 2M NaOH. Absorbance relative to blank wells was read at 405 nm on a Spectromax 240 automated plate reader (Molecular Devices, Sunnyvale, CA, USA). Optical densities (OD) were calculated by mean differences between ELISA test and control wells. To account for inter-assay variation, all OD were converted to arbitrary standard units (SU) based on a standard curve generated in ReaderFit Online Version (Hitachi Solutions America) with a four-parameter log–logit equation. For parametric statistics, anti-hsp IgG SU results were log (SU + 0.01) transformed, now referred to as logSU.

Statistical analysis

Using general linear regression in Genstat version 14, c-hsp60 antibody levels (logSU, n = 94, d.f. = 2) were compared among different age and health (infection status) groups. Following this, MHC variants were evaluated with age class and health status for statistical association with logSU. Only MHC variants found in 10%–90% of koalas studied were analysed, and non-significant MHC variants were dropped from the linear regression by a stepwise procedure (p < 0.05). All diagnostic plots of residuals were checked for normal distribution in Genstat. We systematically used R (R Development Core Team, 2012) for Fisher’s exact test to compare variant presence with (i) age class, (ii) chlamydial disease (“past infected” and “healthy”), and (iii) c-hsp60 antibody seropositivity: “healthy” koalas in the AY age class were used to determine a limit for c-hsp60 antibody seropositivity (two standard-deviations above the mean logSU) of 1.785, whereby all koalas were classified as c-hsp60 antibody seropositive if logSU ≥ 1.785. In order to be inclusive when screening for potential genetic candidates for future, more targeted studies, a less conservative modified false discovery rate (FDR) method (Benjamini & Yekutieli, 2001; Narum, 2006) was used to determine critical significance values.

Results

MHCII DAB and DBB variants

From the 94 koalas, a total of eight DAB (DAB*10, 15, 18, 19, 21–24, GenBank accession numbers JX109927, JX109928, JX109929, JX109930, JX109931, JX109932, JX109933, JX109934) and four DBB variants (DBB*01–04, GenBank accession numbers JX109922, JX109923, JX109924, JX109925) were identified, all of which have been described previously in this population (Lau et al., 2014). Also consistent with Lau et al. (2014), koalas had four to five DAB variants and one to two DBB variants identified per individual, and the DAB*19 and DAB*21 variants were found in a majority of koalas (96.8% and 98.9%, respectively), and were therefore not included in analyses. Although three new “genotype patterns” were identified from OSCP analyses, sequencing of these patterns showed that they comprised new combinations of previously identified MHCII variants.

Association of c-hsp60 antibody levels with age class and chlamydial infection

Both age class and evidence of chlamydial infection were significantly associated with c-hsp60 antibody levels (general linear regression). Koalas over five years of age had significantly higher c-hsp60 antibody titres (logSU predicted mean ± standard error; AM: 1.20 ± 0.28, AA: 0.97 ± 0.31) than young adults aged between two and five years old (−0.77 ± 0.42) (p = 0.001, least significant difference at 5% level (l.s.d.) = 1.03) (Fig. 1). “Past infected” koalas had significantly higher mean logSU values (0.94 ± 0.20) than “healthy” koalas (−0.16 ± 0.41) (p = 0.019, l.s.d. = 0.90) (Fig. 1). Age class and evidence of chlamydial infection were, therefore, incorporated into regression analysis for assessing MHCII variants as predictors.

Figure 1 Mean chlamydial heat-shock protein 60 antibody levels in koalas of three health groups: unknown (U), “healthy” (H) and “past infected” koalas (I) and age classes: aged adult (AA), mature adult (AM), young adult (AY) and unknown (AU).

MHCII variants associated with chlamydial infection, c-hsp60 antibody levels and age

Of the four PhciDAB and four PhciDBB variants found in 10–90% of koalas studied, three were associated, each with a different aspect of the pathogenesis of chlamydial disease or health in the koala: DBB*04 with antibody production, DAB*10 with infection and persistence, and DBB*02 with age. For each group of koalas with the same MHCII variant, Fisher’s exact test found no differences in the proportion of c-hsp60 antibody seropositive (logSU > 1.785) and seronegative (logSU < 1.785) individuals (p = 0.158–1.000, Supplemental Information A). However, general linear regression analysis showed that DBB*04-positive koalas (1.74 ± 0.41, n = 14) had significantly higher mean c-hsp60 antibody (logSU ± s.e.) values than DBB*04-negative koalas (0.52 ± 0.17, n = 80, p = 0.008, l.s.d = 0.88) (Fig. 2, Supplemental Information B).

Figure 2 Mean log anti-chlamydial heat shock protein 60 antibody levels of koalas with (+, blue) and without (−, red) DAB and DBB variants that occurred at between 10% and 90% prevalence in the population studied.

Association of the other two MHCII variants with infection (DAB*10) and age (DBB*02) were identified using Fisher’s exact tests. Based on modified FDR, the critical value for the total number of hypothesis tests performed (k = 8) is p = 0.018 (Benjamini & Yekutieli, 2001; Narum, 2006). On this basis, there was a strong trend for a greater percentage of “past-infected” koalas (87.3%) to be DAB*10-positive than “healthy” koalas (60.0%) (p = 0.021, Table 1). When comparing the three age classes, the proportion of DAB*15 and DBB*02-positive koalas appeared to increase with age, and DBB*02 closely approached the critical significance value (p = 0.019); prevalence of the DBB*02 variant increased from 73.3% of koalas below five years of age, to 87.9% of mature aged koalas, to 100% of koalas above ten years of age (Table 2).

Table 1 Frequency of DAB and DBB variants in “healthy” koalas and “past infected” koalas.

Variant	Proportion “healthy”
(%)	Proportion “past infected”
(%)	Fisher’s exact
p-value	
DAB*10	9/15 (60.0%)	55/63 (87.3%)	0.023	
DAB*15	3/15 (20.0%)	10/63 (15.9%)	0.707	
DAB*22	4/15 (26.7%)	14/63 (22.2%)	0.739	
DAB*24	6/15 (40.0%)	14/63 (22.2%)	0.192	
DBB*01	1/16 (6.3%)	8/63 (12.7%)	0.678	
DBB*02	13/16 (81.3%)	56/63 (88.9%)	0.425	
DBB*03	6/16 (37.5%)	21/63 (33.3%)	0.774	
DBB*04	4/16 (25.0%)	7/63 (11.1%)	0.219	
Notes.

Fisher’s exact p-values close to the critical significance value of p = 0.019 are in bold.

Table 2 Frequency of DAB and DBB variants in koalas in each of three age classes.

Variant	Proportion AY
(%)	Proportion AM
(%)	Proportion AA
(%)	Fisher’s exact
p-value	
DAB*10	12/15 (80.0%)	25/33 (75.8%)	21/25 (84.0%)	0.812	
DAB*15	1/15 (6.7%)	3/33 (9.1%)	7/25 (28.0%)	0.130	
DAB*22	3/15 (20.0%)	8/33 (24.2%)	7/25 (28.0%)	0.883	
DAB*24	1/15 (6.7%)	9/33 (27.3%)	7/25 (28.0%)	0.265	
DBB*01	1/15 (6.7%)	5/33 (15.2%)	2/25 (8.0%)	0.701	
DBB*02	11/15 (73.3%)	29/33 (87.9%)	25/25 (100%)	0.019	
DBB*02
homozygote	7/15 (46.7%)	20/33 (60.6%)	11/25 (44.0%)	0.442	
DBB*03	6/15 (40.0%)	10/33 (30.3%)	8/25 (32.0%)	0.765	
DBB*04	4/15 (26.7%)	1/33 (3.0%)	4/25 (16.0%)	0.048	
Notes.

Fisher’s exact p-values close to the critical significance value of p = 0.019 are in bold.

AY young adult

AM mature adult

AA aged adult

Discussion

Studies in mice and humans have indicated that antibody responses to chlamydial-hsp60 are genetically predisposed (Gaur et al., 1999; Zhong & Brunham, 1992), but this is the first study to investigate associations between MHC and c-hsp60 antibodies and chlamydial disease in a free-living wildlife species, the koala. We identified three candidate MHCII variants; each associated with one of infection (DAB*10), serologic response (DBB*04), or age of presentation (DBB*02). This is consistent with the complex pathogenesis of chlamydial disease and, indeed, parallels the identification of different alleles associated with specific aspects of chlamydial disease in women (Betsou, Sueur & Orfila, 1999; Cohen et al., 2000; Gaur et al., 1999; Kinnunen et al., 2002).

The identification of a higher proportion of Chlamydia-infected koalas carrying the DAB*10 variant, relative to “healthy” koalas, is similar to studies in women where a positive association was identified between specific DQ alleles and C. trachomatis infection or chlamydial cervicitis (Cohen et al., 2000; Ness et al., 2004). If DAB*10 is a susceptibility variant for infection, it may act by displacing another protective variant, such as the phylogenetically similar DAB*15. Although there is only a single amino acid substitution between these two variants, minor changes in MHC amino acid sequences can alter the pattern of epitopes that are presented (Bill et al., 2005; Mealey et al., 2006), thus preventing successful binding to antigenic peptides. Candidates for these peptides might include epitopes of the major outer membrane protein (MOMP), which is required for cellular adhesion and for which particular epitopes are recognised to elicit the Th1 response important for protective immunity against chlamydial organisms (Girjes et al., 1993; Su et al., 1990). MHC restricted epitope recognition in chlamydial infection has been documented in mice, whereby a specific C. trachomatis MOMP Th cell epitope was not recognised by all MHCII haplotypes (Su & Caldwell, 1992). If the Th1 lymphocyte response and the associated IFNγ secretion is limited in infected DAB*10-positive koalas, it may result in the failure to eliminate the organism (Morrison & Caldwell, 2002). Further elucidation of the role of DAB*10 could include the investigation of antigenic peptides recognised by the adaptive immunity of DAB*10-positive and -negative koalas.

A positive association between an MHCII variant (DBB*04) and c-hsp60 antibody levels suggests that this variant recognises and binds particularly well to c-hsp60 epitopes to trigger a strong c-hsp specific humoral immune response. Whether DBB*04 is associated with pathogenesis is less clear, and investigation of this would need to follow several possible mechanisms. It would be interesting to determine whether this variant triggers a cellular response, which might induce type IV-associated pathology, or whether it favours a Th2 response, which might induce pathogenesis by allowing chlamydial persistence. Whether DBB*04-positive koalas produce a greater cellular immune response to hsp60 might be investigated using in vitro assays, similar to those used by Kinnunen et al. (2002) to identify in women associations between lymphoproliferative response to chlamydial hsp60 and HLA class II variants. In addition, elucidation of the Th1/Th2 balance could be gained from the study of the cytokine profile of these koalas (Maher et al., 2014).

If the apparent increase in proportion of DBB*02-positive koalas with age is representative of a trend in the source population, it might be due to enhanced survival of animals with DBB*02, while the opposite would be indicated if this variant is only age-biased in the hospitalised population. While young koalas are admitted for conditions predominantly associated with dispersal, aged koalas are much more commonly admitted with chlamydial disease (Griffith et al., 2013) and this was evident in animals in this study. While DBB*02 is not likely to be a susceptibility variant due to its high prevalence in the population, studies in other species have identified a single susceptibility allele (variant) that is masked by other alleles in heterozygotes (Oliver & Piertney, 2012; Worley et al., 2010) and this might permit the persistence of a detrimental allele. There is no evidence to support this in the current study, as there was no change in the proportion of DBB*02 homozygotes with age (Table 2). It would be of interest to examine the relationship between variant prevalence and age in the source (free-ranging) population.

Since MHC regions generally have high linkage disequilibrium and can disperse across the genome (Belov et al., 2006; Siddle et al., 2009), we cannot rule out the possibility that the associations found in this study instead relate to a gene linked to these MHCII candidate variants. Further characterisation of the koala genome and the genetic location of the MHCII candidate genes would provide insights into this. Due to the complex and multi-factorial nature of wildlife disease, and difficulty in accessing and assessing free-ranging animals, studies relating risk factors to disease in wildlife are very challenging. The presence of archived material was a unique opportunity to begin investigations of the significance of MHCII genetics on the most important infectious disease of koalas but, as with most retrospective studies, had some limitations. Greater consistency in detecting and speciating chlamydial infection, such as with species-specific real-time PCR (Govendir et al., 2012), and more consistent clinical classification of animals, would be desirable and possible with a prospective study. Chlamydia is an extremely challenging disease for this type of study, due to its chronicity, the cryptic nature of some of its pathological changes, and its complex pathogenesis; and the cross-sectional sampling of hospitalised koalas, which are biased for age and disease (Griffith et al., 2013), rather than longitudinal sampling of a free-ranging population, limits some conclusions. Incorporation of MHCII genetics in a longitudinal epidemiological study of a free-ranging population would be ideal; to remove biases, provide a more thorough history of disease progression, and permit more detailed survival studies. Extending studies on a wider spatial scale would also be important, to investigate population-specific variants, and the significance of MHCII to disease in the southern state of Victoria where most populations sampled to date are monomorphic for DBB*02 (Lau et al., 2014). Studies on a wider temporal and spatial scale will require high throughput genotyping methods and the present study provides the basis for a targeted approach to genotyping, for example through variant-specific primers based on the candidate variants we have identified.

Conclusion

In this study we have identified three MHCII variants in the koala that are associated with either chlamydial hsp60 antibody levels (DBB*04), chlamydial infection (DAB*10), or age (DBB*02), providing further support for a significant role of MHCII in the koala, and providing candidates for more focused prospective studies in the future. Stemming from this study, many questions are raised about the complex interaction between MHC and chlamydiosis, including the genetic impact on the Th1/Th2 balance, recognition of specific chlamydial antigens, and resistance or susceptibility to disease. This work provides the foundation and impetus for future work through further characterisation of the role of these candidate variants.

Supplemental Information

Supplemental Information Supplemental Information A shows Fisher’s exact test comparing proportion of MHCII variant-positive c-hsp60 antibody seropositive and seronegative koalas. Supplemental Information B tabulates mean c-hsp60 antibody levels (logSU) associated with specific MHCII variants, which are also presented graphically in Fig. 2.

Click here for additional data file.

We thank the Koala Hospital (Port Macquarie) staff and volunteers, especially Cheyne Flanagan, for assistance with sample collection, and Peter Thomson for assistance with statistical analyses.

Additional Information and Declarations

Competing Interests

Author Contributions

Animal Ethics

The authors declare there are no competing interests.

Quintin Lau conceived and designed the experiments, performed the experiments, analyzed the data, contributed reagents/materials/analysis tools, wrote the paper, prepared figures and/or tables, reviewed drafts of the paper.

Joanna E. Griffith and Damien P. Higgins contributed reagents/materials/analysis tools, reviewed drafts of the paper.

The following information was supplied relating to ethical approvals (i.e., approving body and any reference numbers):

The University of Sydney Animal Ethics Committee: AEC N00/5-2009/1/4829

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
