# Peer review of "Identification of MHCII variants associated with chlamydial disease in the koala (Phascolarctos cinereus)"

_PeerJ, doi:10.7717/peerj.443_

## Round 0.1 · original submission · Major Revisions

· Academic Editor

Major Revisions

The reviewers' comments are constructive, so I hope that you can revise the manuscript accordingly.

Reviewer 1 ·

Basic reporting

The introduction is well written, informative and states the aims and puts them into the context of the general field of research. In the results section and discussion, however, readability could be slightly increased by being more specific (e.g. by naming the variants that are associated with the different aspects of disease instead of saying ‘one variant was associated with..’). Figures could be made slightly clearer by e.g. labelling the Y axis ‘hsp AB logSU...’ (or even 'hsp AB titre'?) If numbers in tables are highlighted, this should be explained in the caption.

Experimental design

Accession numbers for the sequenced variants should be given.

Validity of the findings

The respective statistical test used for each result should be mentioned in the text.
Sometimes it is not clear to me, to which tests the text refers to (see more details under 'general comments'). Which test was e.g. used in the table presented under 'Supplentary material B'? If multiple statistical comparisons where made, was this always corrected for?

Additional comments

Line 22: Please clarify to which proteins you refer here.
Lines 78-81: Where does this DNA come from? For how many individuals swabs where available? How many necropsies? Please clarify methods and sample numbers.
Line 150: Does this mean the residuals were normally distributed?
Line 153: Fisher’s exact test was not published by ‘R Development Core Team 2012’ – please correct.
Line 155: Aims should be introduced in the introduction, not under ‘statistical analysis’.
Line 156 (and elsewhere in the text, figures and tables): Please specify that you mean ‘c-hsp antibody (AB) seropositivity’.
Line 166: It would be more informative to say ‘sequencing’ instead of ‘characterisation’ (if that is how the variants were identified).
Line 169: Delete ‘as expected’, results should be presented without interpretation.
Lines 180f (and elsewhere in the text): Please name the respective genotypes instead of saying ‘one was associated’. It is not clear to which results you are referring in this sentence.
L 181: This sentence is confusing:’Although no MHCII variants were associated with seropositivity …, DBB*04-positive koalas…had significantly higher mean c-hsp60 antibody values…’. Was there an association or not? Did you test the same question through different tests? If you tested each variant in a single test, corrections for multiple comparisons might be necessary.

Table 1: Why are there different numbers of ‘healthy’ individuals for DAB (N=15) and DBB (N=16)?

Reviewer 2 ·

Basic reporting

The paper by Lau et al examines associations between MHC class II variants and antibody responses to c-hsp60 with Chlamydia in wild koalas. This study used archival tissue samples from koala hospital admissions. Although this study has some limitations due to the nature of sample collection from the koala, it does provide preliminary information and indications for the direction of future research.
Figure 1 should include the numbers of individuals in each cohort identified.
An indication of the natural lifespan of koalas would be helpful in understanding the prevalence of infection in each of the age cohorts.

Experimental design

The authors are aware that the experimental design is not ideal but this study still has value in its current form.

Validity of the findings

No comments

Additional comments

See above

---

## Round 0.2 · accepted · Accept

· Academic Editor

Accept

Please go through the manuscript for any typos and errors.